# Evidence-Based Pharmaceutical Care in Community Pharmacies: A Survey of 595 French Pharmacists

**DOI:** 10.3390/pharmacy11050161

**Published:** 2023-10-07

**Authors:** Louis Bosson, Francesca Veyer, Jean-Didier Bardet, Céline Vermorel, Alison Foote, Benoit Allenet, Jean-Luc Bosson

**Affiliations:** 1Pharmacie Sainte Croix, 26200 Montélmar, France; louis.bosson2@gmail.com; 2Pharmacie Duverney Joux, 38600 Fontaine, France; francesca.veyer@gmail.com; 3UFR de Pharmacie, University Grenoble-Alpes, 38700 La Tronche, France; jean-didier.bardet@univ-grenoble-alpes.fr (J.-D.B.); ballenet@chu-grenoble.fr (B.A.); 4TIMC, CNRS UMR5525, University Grenoble-Alpes, 38400 Saint Martin d’Hères, France; celine.vermorel@univ-grenoble-alpes.fr; 5UF Pharmacie Clinique, Pole Pharmacie, CHU Grenoble-Alpes, 38043 Grenoble, France; 6Public Health Department, CHU Grenoble Alpes, 38043 Grenoble, France

**Keywords:** evidence-based practice, pharmaceutical care, community pharmacy, France, over-the-counter medication, sources of information

## Abstract

We assessed the use of evidence-based practice (EBP) among pharmacists working in community pharmacies in France and the factors linked to this practice. During 3 months in 2018, an online survey was sent to over 7000 active pharmacists and posted on pharmacists’ social media sites. In total, 595 pharmacists completed the questionnaire. The responders were on average younger than the general population of community pharmacists. The 40-item questionnaire described four fictional clinical cases reflecting typical situations (conventional medicine and complementary and alternative medicine) encountered daily by community pharmacists. Multiple-choice responses were proposed and scored according to whether they reflected EBP. A high total score indicated behaviour in line with EBP. We observed 344/595 participants with a positive EBP score (57.8% [53.7–61.8%]). Univariate and multivariate analyses were used to evaluate factors that might explain adherence to EPB (the pharmacy’s characteristics, the pharmacist’s status, the mode of continuing education and sources of information). The majority relied on pharmaceutical industry and other biased and/or non-evidence-based sources, particularly concerning information on homeopathic products. The consultation of independent reviews, health agency recommendations and peer-reviewed scientific journals was associated with evidence-based decisions. In contrast, reliance on pharmaceutical industry documents, personal experience and informal handbooks was linked to lower EBP scores. The level of EBP use by French community pharmacists needs to be improved to ensure that good-quality, science-based advice is given to customers.

## 1. Introduction

Nowadays, community pharmacists (CPs) are considered to be healthcare providers [1]. In Europe, pharmaceutical care is defined as “the pharmacist’s contribution to the care of individuals in order to optimize medicines use and improve health outcomes” [2,3]. Changes in the role of pharmacies in the community underlie this description of the pharmacist’s function. Day-to-day pharmacy practice is no longer restricted to the dispensation of medicines. Professional pharmacy services currently include vaccinations, triage services, patient medication reviews and pharmaceutical planning [4]. Over time, for many patients, the CP has become their first contact with the healthcare system. In France, less restricted access to medicines has made this possible. Indeed, the offer of over-the-counter (OTC) medications has expanded and now includes the dispensing of drugs previously available only on prescription. The CP is also perceived as a privileged interlocutor for complementary and alternative medicines (CAMs) (with around 100 h of initial training devoted to CAMs), making a bridge between conventional medicines and natural health products [5]. Finally, changes in French legislation over the past 10 years have changed the CP from a simple retailer to a pharmaceutical services provider [1], not only for patients but also for other healthcare professionals, including physicians. The CP has become a co-decision-maker, especially when working in an integrated-care environment.

“With greater power comes greater responsibility” [6] could be the statement describing this change in function that ranges from the management of minor ailments to the provision of pharmaceutical care for patients with chronic conditions [7]. Responsible practice implies that it is evidence-based practice (EBP) [8]. The adoption by CPs of an approach based on EBP has been studied in terms of perceptions [9,10,11,12], the evaluation of professional practices [13,14,15], the evaluation of interventions during initial training [16,17] or in everyday practice [18,19].

However, the studies that have evaluated EBP in the community pharmacy setting have mainly focused on OTC drugs. In contrast, the primary objective of the present study was to assess attitudes towards and the application of EBP, whatever the nature of the health product (OTC medication, CAMs or prescription-only medicines). Our study is a survey of community pharmacists to assess their level of adherence to the concept of EBP. In particular, we wanted to test whether recent developments in the pharmacy curriculum, with the introduction of the EBP concept in initial training (for example, the introduction, over 10 years ago, of lectures on EBP at the Grenoble Faculty of Pharmacy), are reflected in everyday practice. We also asked whether facilitated access to independent sources of knowledge (through the Internet, for example) had increased the awareness and practice of EBP. We used an original methodology based on short fictional descriptions of typical concrete cases followed by multiple-choice questions.

## 2. Materials and Methods

A cross-sectional study was carried out among CPs and 6th (final)-year pharmacy students practicing in a community pharmacy or studying in France. In France, final-year pharmacy students taking the CP specialty undertake an internship of 6 months in a community pharmacy. Pharmacy technicians and students at earlier stages in their pharmacy studies were not included. The 40-item online questionnaire (Appendix A) was sent between 1 April and 1 June 2018 using a public email database containing more than 7000 different contacts of CPs. The questionnaire was also posted on five Facebook groups (restricted to CPs and pharmacy students). Participation in the study was voluntary. Returning a completed questionnaire implied the responder’s consent to participate in the survey.

The questionnaire had three parts, as follows: In part A, four different fictional clinical cases were presented, each with four possible answers. The respondents were told that there was no right or wrong answer, and they were asked to tick the answer that was closest to what they would do in their everyday practice. Part B consisted of questions on the pharmacist’s status, whether they attended any continuing education and the sources of information they usually consulted concerning allopathic medicines and CAM; Part C was questions about socio-demographics and the characteristics of the pharmacy where they worked, with an optional open response section for any remarks they would like to make. 

The primary objective of this study concerned the multiple-choice part of the questionnaire (Part A) that presented four fictional cases typical of those encountered in the daily practice of community pharmacists. The topics were the replacement of the influenza vaccination by a homeopathic product, a dietary supplement (red yeast rice) as a substitute for statins, the use of the herbal preparation St. John’s Wort for moderate depression and the choice of a nonsteroidal anti-inflammatory drug (NSAID) (Appendix A). For each case, the respondents had to choose one of the four proposed responses that best corresponded to their current professional practice. 

To give the respondents confidence and to encourage them to complete and return the questionnaire, for one case (n° 2–St John’s Wort), all the answers proposed corresponded to good EBP. This was also to show that the questionnaire was not intended to judge a CAM if its use was backed up by scientific evidence. The pharmacists’ choices of response to this case were not analysed.

A pilot version of the questionnaire had been tested by 11 community pharmacists, physicians, university lecturers and hospital pharmacists and 30 pharmacy students. 

The primary outcome was the total score from the multiple-choice responses to the clinical cases. For each case, 4 points were attributed for the response closest to EBP, 3 points if the respondent knew the best EBP response but felt they could not say that to the patient, 2 points if the respondent did not know the appropriate EBP answer, and 1 point if the response was counter to EBP. Thus, the minimum total score was 3 and the maximum score was 12. The ranking attributed to responses had been justified by a review of the literature that included the Cochrane Database of Systematic Reviews, the French “La Revue Prescrire” (a French independent publication on professional practice and drugs, produced by a network of volunteer experts) and the recommendations of the French health authorities and/or professional associations if necessary (Appendix A) [20,21,22,23,24,25,26,27,28,29,30,31]. Based on the distribution of multiple-choice scores, a threshold was determined such that respondents could be categorized into two groups, EBP positive (high score) and EBP negative (low score). 

The second part of the questionnaire collected data on the type of pharmacy where the CP worked and the pharmacist’s status, whether they attended any continuing education and the sources of information they usually consulted. 

Analysis: We present qualitative variables using number and frequency and mean and standard deviation for continuous variables. We calculated the proportion of EBP-positive participants (with a 95% confidence interval) using the binomial exact method. We used one-sample tests of proportions or means to assess how representative the survey respondents were of the general population of practicing CP in France (age, sex and region). To compare modes of continuing education according to the pharmacist’s status, we used the Chi2 test. Univariate tests were conducted to evaluate the association between responses and the profile of the participants and the pharmacies where they worked. A multivariate analysis with logistic regression was conducted using variables selected from the univariate analyses (having a *p* ≤ 0.20) that might explain the adherence of participants to EBP (EPB+, or EBP−). The final model was selected after a manual step-down selection procedure. An a priori alpha of less than 0.05 was considered as significant. Stata 15 software for OSX was used for the data analyses. 

The statistical analysis and content of this article are consistent with STROBE international recommendations to report cross-sectional studies (if applicable).

## 3. Results

Out of the 599 pharmacists and sixth-year pharmacy students who returned the survey, four questionnaires were incomplete. Finally, 595 completed questionnaires were included in the database for statistical analysis. The mean age of the respondents was 38 ± 11.7 years, and they came from all 13 metropolitan regions of France, as well as the overseas territories (Table 1). On average, the respondents to the survey were younger than the general population of CPs in France (Table 1). Given the use of Facebook groups for which the number of pharmacists who accessed the site but did not respond to the questionnaire was not known, the response rate could not be calculated.

Table 2 shows the distribution of pharmacists’ responses to the cases presented in the questionnaire (except case 2, which was not scored as all responses were good EBP), and Figure 1 shows the distribution of the total score, supporting the categorization of responders into two groups with a threshold between 7 and 8 (negative EBP: score ≤ 7; and positive EBP: score ≥ 8). We observed 344/595 participants with a positive EBP score (57.8% [53.7–61.8%]).

French pharmacists use a variety of continuing education methods (both self-instruction or following organized courses) to increase their knowledge and keep up to date with developments concerning both conventional medications, devices, etc., and CAMs (Table 3). 

After qualifying as a pharmacist in France (Pharm D: 6 years of pharmaceutical studies plus an internship), the most frequent post-graduate courses taken by CPs were orthopaedics (mandatory diploma to counsel for medical devices in orthopaedics) (48.6%) (289/595); homeopathy (these diploma courses were recently withdrawn in public sector schools of pharmacy) (8%) (48/595); and home care (diploma in helping dependent patients remain in their homes) (7%) (42/595). Many participants had not completed any post-graduate diploma course. 

Concerning the topics broached in the cases presented in the questionnaire (a homeopathic product instead of influenza vaccination, dietary supplements to replace statins and CAM instead of nonsteroidal anti-inflammatory drugs (NSAIDs)), the most commonly consulted sources of information are shown in Table 4.

The logistic regression selected the variables that were positively or negatively associated with the group using EBP (EBP positive) or the group ignoring EBP ((EBP negative). An odds ratio (OR) > 1 was positively correlated with the proper use of EBP, and an OR < 1 was negatively correlated with the good use of EBP. The factors positively correlated with the use of EBP were being a junior CP or final-year student (compared to being a pharmacy owner (*p* < 0.01) and frequent use of health agency recommendations for guidance *p* < 0.01) as well as reading ”*Prescrire*” (*p* = 0.010), compared to not consulting them. The factors negatively correlated with the proper use of EBP were the frequent use of information provided by pharmaceutical laboratory sales representatives (*p* = 0.012), relying on personal experience (*p* = 0.016) and/or having a diploma in homeopathy (*p* = 0.024) (Table 5). The percentage of EBP+ according to age group was very highly correlated with the CP’s status in the pharmacy (owner, salaried, substitute or junior CP/final-year student) (Appendix A).

## 4. Discussion

The choice of responses to a questionnaire that presented different clinical cases gave us some insights into the knowledge and use of EBP by community pharmacists and associated factors such as their status in the pharmacy, the sources of information they usually use and whether they pursued any continuing education. Most previous surveys focusing on EBP in community pharmacies used qualitative methods, questionnaires with multiple open-text answers or ordered and Likert scales [10,11,12,14]. 

More than 80% of questionnaire respondents checked the influenza vaccine as the first-line measure for influenza prevention. However, 12% affirmed that they would advise replacing the influenza vaccine by the homeopathic product “Influenzinum” (a homeopathic product in the form of granules containing a dilution of the current year’s influenza vaccine), and considered it as an effective option, despite no scientific proof. These answers need to be to be considered in the context of French attitudes towards the influenza vaccine and homeopathy at the time the questionnaire was distributed (2018). The authorization for community pharmacists to perform influenza vaccinations was introduced in 2018. Likewise, homeopathy has been reimbursed by the French national health insurance scheme for decades. However, its place in drug therapy and its reimbursement by French social security has been criticized for several years by both the pharmaceutical and medical professions. This controversy finally led most French universities to discontinue their university diploma courses in homeopathy. Moreover, the study shows that pharmacists who hold a university diploma in homeopathy are more inclined to make decisions that are not supported by EBP. Thus, the impact of homeopathy courses does not seem to be confined to the sole practice of homeopathy itself. Perhaps the fact of undergoing training in homeopathy reflects adherence to a mode of reasoning based on beliefs (or theories) without taking into account scientific evidence.

The survey showed that some pharmacists are directed to non-EBP information sources in courses run by the pharmaceutical industry, rely on their personal experience and consult informal non-peer reviewed handbooks (often provided by sales representatives and available in the dispensary). Indeed, Hanna et al. [11] suggested that decisions taken regarding OTC medications and CAMs are mostly the result of personal experience (individual, familial or patient feedback). Furthermore, it is interesting to note that the product’s safety seems more important than the product’s efficacy for some pharmacists [11,13,14,15]. In view of the results of qualitative studies on OTC sales conducted by Hanna [11] and Rutter [13], when in doubt and faced with commercial pressure, some pharmacists choose to secure the sale rather than giving EBP advice. However, in the present study, for some pharmacists, an alternative stance was seen in all three cases. This consisted of giving the patient the non-EBP product while at the same time recognizing their action was not EBP. 

The case concerning red yeast rice is interesting because it concerns a CAM product containing a phytotherapeutic molecule identical to lovastatin (monacolin K). This active agent in red yeast rice has been assessed according to EBP criteria; however, the few randomized studies were relatively small in size [32]. The monacolin K concentration varies according to how the red rice is prepared and can sometimes contain traces of toxic substances [33]. The survey showed that nearly half of the respondents would deliver red yeast rice to reduce cardiovascular risk in patients who experienced adverse muscular side effects with a statin and had to discontinue it. While it seems that the pharmacist is making a therapeutic choice based on patient safety, the answers to this clinical case raise questions as to how pharmacists are trained to evaluate the benefit/risk balance. 

The survey revealed that the majority of pharmacists still use information sources that are subject to bias. Indeed, the majority of sources used by community pharmacists for documentation on CAM are subject to bias (training from the pharmaceutical industry, personal experience and informal CAM handbooks). This is possibly because high-quality randomized trials of CAM are difficult to fund and often have negative results, making them difficult to publish (publication bias). These broad figures (except on CAM) are comparable to those from Mamiya et al. [34] in another international setting: concerning tools for continuing self-development, 34.8% of respondents answered “at home using books purchased by themselves and web searches”, 24.2% “study meeting of medical associations such as Japanese Society of Hospital Pharmacists or Japan Pharmaceutical Association” and 10.8% respondents answered “pharmaceutical companies (wholesalers) study meetings”. It should be noted that some pharmacists are not at ease reading information in English, and this might make it more difficult to access reports of high-quality studies published in international journals. Furthermore, it appears that some respondents rely on information from the regulatory authorities but are often not up to date with changes, because they rely on the package inserts only. The results of the multivariate analysis confirm the importance of the source of information used in the choices made, with the use of sources less subject to bias, such as the French review “Prescrire” (the only independent source of information on drugs in French, funded entirely by its subscribers), being associated with science-based decisions. The reliance on personal experience and pharmaceutical industry-related training was correlated with decisions not supported by scientific evidence. The results also suggest that making science-based decisions is more frequent among junior pharmacists who use multiple and varied sources of information, unlike more senior pharmacists with more years of experience, but who are possibly less inclined or have less time to explore news sources. This might also be explained by recent changes in the pharmacy curriculum. Initial training now incorporates teaching modules aimed at developing an EPB culture among pharmacy students. This is particularly the case for teaching modules concerning the critical reading of articles or those in clinical pharmacy. The study was transversal, so conclusions regarding either a change in attitude according to generation (age), or to a phenomenon of erosion during professional exercise cannot be advanced. 

As a specification of evidence-based medicine, evidence-based practice in pharmacy relates to the promotion of “judicious, appropriate and safe use of medicines” [35]. One of the key issues is education. One of the most important determinants of adult learning is its relevance to clinical practice [36]. Teaching healthcare professionals to discern the most relevant and independent information source so that they can use it in their practice is essential. It also appears necessary to integrate lessons based not only on EBP but also on applying the principals of EBP to clinical practice in all continuing education courses. It seems unrealistic to expect French community pharmacists to routinely search and critically appraise clinical trial reports in the international literature, particularly due to the language barrier. This is why the regular consultation of up-to-date, simplified review articles in independent publications such as “Prescrire” should be encouraged.

The EBP approach is still at its beginning among French community pharmacists. Even if our methodology is open to improvement (we do not have a gold standard for evaluating this EBP approach among pharmacists), our study highlights some encouraging signs. Nearly 60% of pharmacists surveyed have adopted an EBP-compatible approach, a figure probably overestimated given the over-representation of young pharmacists in our sample. Indeed, CPs who had only recently completed their studies would have been introduced to the concepts of EBP. However, more interesting is the qualification of this positioning, particularly a greater consideration of evidence in decision making on the part of young pharmacists, combined with greater use of scientifically valid sources of information (recommendations for practice such as those in the “*Prescrire*” journal). 

These results should be confirmed in a larger study. However, in the meantime, one could hypothesize that teaching pharmacy students simple methods to access non-biased sources of information can lead to better EBP. This should also be a challenge for practicing pharmacists, who need to keep their knowledge and skills up to date instead of working “from their own experience”.

There are several potential biases and limitations to our study, particularly concerning the representativeness of the sample. The large number of respondents provides a broad overview of the practices of community pharmacists practicing in France. Data collection was carried out via an online questionnaire that had been tested by several community pharmacists, university hospital practitioners and pharmacy students to ensure its coherence, feasibility and quality. Our sample is slightly younger than that of French pharmacists as a whole, which can be explained by our decision to include student pharmacists in their end-of-study internship in the sixth year. The percentage of responders who are also pharmacy owners was higher than national figures for practicing CPs. We contacted CPs in all French regions, but for clarity, Table 1 only shows the five regions with the highest proportions of responders and the number of active CPs in these regions. The questionnaire could be completed in the pharmacist’s free time, allowing the respondent to focus on the benefit/risk evaluation. It did not take into consideration the external constraints that usually accompany such decision making (restricted time, patient/customer or managerial pressure, etc.). The self-declarative format of the questionnaire may have induced bias towards respondents giving what they considered to be the desired responses rather than ones reflecting the pharmacist’s actual everyday practice. 

## 5. Conclusions

The data collected from 595 pharmacists showed that nearly 60% apply EBP to their everyday practice. This allowed us to make an overview of evidence-based practice among French community pharmacists and to highlight some factors influencing their choices. Junior community pharmacists tend to be more willing to apply EBP compared to their more senior colleagues. The sources of information mainly consulted by the pharmacist were also linked to their respect of EBP. These results should be confirmed in a larger study. However, in the meantime, one could hypothesize that teaching pharmacy students simple methods to access non-biased sources of information can lead to better EBP. This should also be a challenge for practising pharmacists, who need to keep their knowledge and skills up to date instead of working “from their own experience”.

The two key issues to continuing the EBP trend in community pharmacy practice are to include EBP in the initial and continuing training of pharmacists and to help them access independent, synopsis and practice-oriented sources of information. 

## Figures and Tables

**Figure 1 pharmacy-11-00161-f001:**
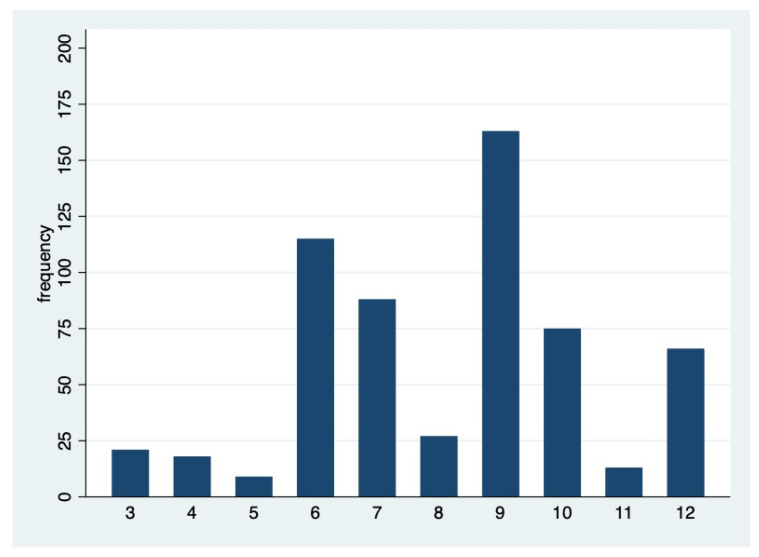
Distribution of the total multiple-choice score (*n* = 595).

**Table 1 pharmacy-11-00161-t001:** Characteristics of respondents and the pharmacies in which they work.

Characteristic	Respondents to Survey (Total = 595)	Registered CPs in France (N = 53,975)	*p*-Value
Male, n (%)	196 (32.9%)	16875 (31.3%)	0.378
Mean age, years (SD)	38.2 (11.7)	46.2	<0.01
Age band, n (%)			
<30 years	193 (32.4%)		
30–39 years	167 (28.1%)		
40–49 years	105 (17.6%)		
≥50 years	130 (21.8%)		
Pharmacist’s status, n (%)			
Pharmacy owner CP	330 (55.5%)	25,189 (46.7%)	<0.01
Salaried CP	176 (29.6%)		
Substitute CP	30 (5.0%)		
Junior CP or final year student	59 (9.9%)		
Pharmacy location, n (%)			
City centre	128 (21.5%)		
Urban area	73 (12.3%)		
Social housing estate	38 (6.4%)		
Rural area	216 (36.3%)		
Suburban area	140 (23.5%)		
Regions of France with most responders, n (%)			
Provence and Cote d’Azur (Mediterranean SE of France)	88 (14.8%)	4480 (8.3%)	<0.01
Auvergne Rhone Alpes (SE France)	76 (12.8%)	6821 (12.6%)	0.921
Aquitaine (SW France)	67 (11.3%)	5375 (10%)	0.289
Loire valley (central France)	59 (9.9%)	2150 (4%)	<0.01
South of France	54 (9.1%)	4650 (8.6%)	0.689

**Table 2 pharmacy-11-00161-t002:** Responses to the 3 clinical cases used to evaluate the use of EBP.

Case	Type of Response	N = 595	Points
	EBP response with action	479 (80.5%)	4
	EBP response without action	35 (5.9%)	3
	Did not know	8 (1.3%)	2
	Not EBP	73 (12.3%)	1
3	Red yeast rice as a substitute for statins, n (%)		
	EBP response with action	278 (46.7%)	4
	EBP response without action	26 (4.4%)	3
	Did not know	25 (4.2%)	2
	Not EBP	266 (44.7%)	1
4	Choice of NSAID, n (%)		
	EBP response	149 (25.0%)	4
	Did not know	196 (32.9%)	2
	Not EBP	250 (42.0%)	1

**Table 3 pharmacy-11-00161-t003:** Mode of continuing education according to pharmacist’s status. (Several answers were possible).

	Junior CP (n = 59)	Salaried CP (n = 176)	Substitute CP (n = 30)	Owner CP (n = 330)	*p*	All (n = 595)
Professional journals ^1^	46 (78.0%)	138 (78.4%)	23 (76.7%)	182 (55.2%)	<0.01	389 (65.4%)
Pharmaceutical industry information ^2^	41 (69.5%)	140 (79.5%)	17 (56.7%)	188 (57.0%)	<0.01	386 (64.9%)
Conferences ^3^	21 (35.6%)	99 (56.3%)	10 (33.3%)	223 (67.6%)	<0.01	353 (59.3%)
University ^4^ courses	28 (47.5%)	75 (42.6%)	19 (63.3%)	105 (31.8%)	<0.01	227 (38.2%)
Personal contacts ^5^	26 (44.1%)	42 (23.9%)	10 (33.3%)	80 (24.2%)	<0.01	158 (26.6%)
Reading “*La revue Prescrire*” ^6^	15 (25.4%)	34 (19.3%)	6 (20.0%)	63 (19.1%)	0.73	118 (19.8%)

^1^ Professional magazines for pharmacists (e.g., Le Moniteur des Pharmacies, Le Quotidien du Pharmacien); ^2^ training materials offered by the pharmaceutical laboratories; ^3^ attending conferences and training offered by organizations specializing in continuing education (e.g., UTIP, etc.); ^4^ continuing university education (university diplomas in various domains); ^5^ regular contact with other healthcare professionals (peer group, home/health centre, etc.); ^6^ “La Revue Prescrire” (or “*Prescrire*”) is a monthly publication of mainly reviews in French that address developments in disease management, medications and medical techniques and technologies. *Prescrire* contains no advertising and is financed by subscriptions. *Prescrire* mainly publishes reviews prepared by its own staff. It is run by a not-for-profit organization and is independent of the pharmaceutical industry.

**Table 4 pharmacy-11-00161-t004:** Sources of information usually used.

A. Sources (Used More than 50% of the Time) on Homeopathic and Herbal Medicines *	N = 595
Handbooks for both pharmacists and the general public	291 (48.9%)
Summary of Product Characteristics (SPC), Vidal, etc.	251 (42.2%)
Training materials provided by representatives of pharmaceutical laboratories	238 (40.0%)
Discussions with colleagues	216 (36.3%)
Professional magazines for pharmacists (e.g., Le Moniteur des pharmacies, Le quotidien du pharmacien)	210 (35.3%)
Personal experience	188 (31.6%)
French Language Pharmacy Learned Society publications	137 (23.0%)
Health agency recommendations (ANSM, HAS, etc.)	130 (21.8%)
La revue Prescrire	73 (12.3%)
Clinical trials found in reliable databases (Pubmed, Cochrane, Uptodate)	51 (8.6%)
B. Sources (used more than 50% of the time) on the best choice of NSAID **	
Summary of Product Characteristics (SPC), Vidal etc	528 (88.7%)
Health agency recommendations (ANSM, HAS, etc.)	265 (44.5%)
French Language Pharmacy Learned Society publications	231 (38.8%)
Professional magazines for pharmacists (e.g., Le Moniteur des pharmacies, Le quotidien du pharmacien)	222 (37.3%)
Discussions with colleagues	178 (29.9%)
La revue Prescrire	113 (19.0%)
Handbooks for both pharmacists and the general public	101 (17.0%)
Personal experience	98 (16.5%)
Clinical trials found in reliable databases (Pubmed, Cochrane, Uptodate)	39 (6.6%)
Training materials provided by representatives of pharmaceutical laboratories	25 (4.2%)

* Cases 1, 2 and 3; ** Case 4.

**Table 5 pharmacy-11-00161-t005:** Univariate and multivariate logistic regression: factors correlated with a positive EBP score (*n* = 344/595).

	Univariate OR [95% CI]	Multivariate OR [95% CI]
Health agency recommendations (ANSM, HAS, etc.) *	1.33 [1.15–1.53] *p* < 0.01	1.25 [1.06–1.47] *p* < 0.01
Personal experience *	0.85 [0.73–0.99] *p* = 0.041	0.82 [0.69–0.96] *p* = 0.016
Reading “La revue Prescrire” *	1.29 [1.13–1.47] *p* < 0.01	1.21 [1.05–1.39] *p* = 0.010
Clinical trials found in reliable databases *	1.29 [1.07–1.56] *p* < 0.01	/
Consulting magazines for pharmacists *	1.23 [1.10–1.38] *p* < 0.01	/
Training materials provided by pharmaceutical laboratories **	0.85 [0.74–0.98] *p* = 0.023	0.83 [0.71–0.96] *p* = 0.012
Number of pharmacists in the dispensary	1.19 [1.03–1.38] *p* = 0.016	/
Having a diploma in homeopathy	0.49 [0.27–0.89] *p* = 0.020	0.47 [0.25–0.90] *p* = 0.024
Being male	1.40 [0.99–1.99] *p* = 0.060	1.53 [1.04–2.25] *p* = 0.030
Pharmacist status		
Owner CP	1.00	1.00
Salaried CP	1.34 [0.93–1.94] *p* = 0.120	1.38 [0.91–2.09] *p* = 0.131
Substitute CP	1.82 [0.82–4.00] *p* = 0.139	1.62 [0.71–3.74] *p* = 0.255
Junior CP/final year student	3.21 [1.67–6.17] *p* < 0.01	3.07 [1.55–6.08] *p* < 0.01

* in clinical case *n*°4 involving NSAIDs; ** for homeopathic and herbal medicine.

## Data Availability

Data are available for academic use on reasonable request to the corresponding author.

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
