# Peer review of "Evidence-Based Pharmaceutical Care in Community Pharmacies: A Survey of 595 French Pharmacists"

_pharmacy, 2023, doi:10.3390/pharmacy11050161_

Round 1
Reviewer 1 Report
For the publication of the manuscript, a series of significant improvements are necessary:
- abstract: several results obtained in the experimental part must be included;
- introduction: an improvement and a more concrete presentation of the purpose of the study with much better argumentation is necessary;
- methodology: it is necessary to better specify the design of the questionnaire used in the study;
- results: in some parts it would be desirable to apply statistical processing (for example to table 1) and it would also be interesting to correlate the data with the age categories;
- some of the conclusions can be moved to the discussions.
Reviewer 2 Report
The article by Bosson and colleagues describes the results of a study that tests community pharmacists' implementation of evidence based practice while making recommendations around complementary therapies. The study is well described and interesting with some valuable results.
The only issue that I have is that the categories that are listed in the original questionnaire under "resources used" for the three scenarios and "your status in the pharmacy" do not correspond with the categories in the article. This needs to be clearly addressed in the methodology. For example, in the questionnaire, the categories for pharmacists are "pharmacy owner", assistant pharmacist", "substitute pharmacist" and "6th year student". In the article they are listed as "pharmacy owner", 'salaries CP", "temporary CP" and "junior CP". For resources, these are even more diverse. Please address in the methods section how this was grouped. Also please check the article for typos. Table 1 seems to have two top rows that are irrelevant.
Round 2
Reviewer 1 Report
I accept in present form